# Continuous Perfusion Experiments on 3D Cell Proliferation in Acoustic Levitation

**DOI:** 10.3390/mi15040436

**Published:** 2024-03-25

**Authors:** Luca Fabiano, Shilpi Pandey, Martin Brischwein, Morteza Hasanzadeh Kafshgari, Oliver Hayden

**Affiliations:** 1Heinz-Nixdorf-Chair of Biomedical Electronics, School of Computation, Information and Technology, Technical University of Munich, TranslaTUM, 80333 Munich, Germany; luca.fabiano@tum.de (L.F.); brischwein@tum.de (M.B.); morteza.kafshgari@tum.de (M.H.K.); 2Central Institute for Translational Cancer Research (TranslaTUM), School of Medicine, Technical University of Munich, 80333 Munich, Germany; 3Department of Radiation Oncology, School of Medicine, Technical University of Munich, 80333 Munich, Germany

**Keywords:** bulk acoustics, 3D cell trap, microfluidics, in vitro cell test

## Abstract

An acoustofluidic trap is used for accurate 3D cell proliferation and cell function analysis in levitation. The prototype trap can be integrated with any microscope setup, allowing continuous perfusion experiments with temperature and flow control under optical inspection. To describe the trap function, we present a mathematical and FEM-based COMSOL model for the acoustic mode that defines the nodal position of trapped objects in the spherical cavity aligned with the microscope field of view and depth of field. Continuous perfusion experiments were conducted in sterile conditions over 55 h with a K562 cell line, allowing for deterministic monitoring. The acoustofluidic platform allows for rational in vitro cell testing imitating in vivo conditions such as cell function tests or cell–cell interactions.

## 1. Introduction

“The true nature of a cancer disease becomes visible when it develops in conjunction with other cells and tissues in the body”, as described in Understanding Cancer [1]. This is the reason why preclinical animal models are indispensable in cancer research. Still, due to human tumors’ genetic and epigenetic heterogeneity and the differences in the immune system, they remain imperfect, leading to biased results [2]. In vitro cell tests and the emerging field of organoids or tumoroids are promising alternatives [3,4]. However, studying the interaction of immune cells with tumor cells remains challenging, as perfusion experiments are time-consuming and complex [5]. It is necessary to develop advanced systems that enable a robust mimicking of the cellular micro-environment in vitro, even with primary cells from patients. Despite major efforts of the community, broad clinical acceptance of in vitro cell models is still low, with mainly niche applications being favored due to costs, throughput, and time-to-results. Amongst others, 2D culture models cannot capture 3D tissue morphology. This is an inadequate situation considering the increasing importance of personalized cancer therapies.

Conventional 3D cell cultures can mimic tumor microenvironments closely, for example, by encapsulating cells in hydrogel-based matrices to provide natural conditions for stem cell-like growth and homotypic cellular interaction [6]. On the other hand, multicellular spheroids can simulate heterotypic cell–cell interactions, e.g., it is known that the spheroid model complexity impacts tumor cell drug responses [7]. Today, tumor organoid cultures are established only for a few cancer types, and the formation of compact spheroids from heterotypic cell suspensions lacks standardization [8]. Further spheroid fabrication methods, such as the hanging drop method, as explained in Jeong et al., 2022, have the disadvantages of low-throughput, long time expenditure, a complicated procedure, and large heterogeneity in spheroid size [9]. Despite the significant efforts of the community, broad clinical acceptance of in vitro cell models is still low, with mainly niche applications limited by costs, throughput, validity, and time-to-results [10].

Developed by Nobel Prize winner Arthur Ashkin, optical tweezers use focused laser beams to manipulate and move physical objects on a microscopic level [11]. This approach has found numerous research applications, but the high photon density can lead to photodamage, and low trapping forces limit the opportunities for perfusion experiments [12]. Contrary to optical tweezers, acoustic tweezers emerged for high trap forces in the nN regime, allowing perfusion conditions and the manipulation of cell aggregates. For workflow solutions using only minute amounts of patient samples, microfluidic principles can be applied to minimize the dead volume and fully exploit the various opportunities for perfusion experiments for cell–cell interaction studies. Today, microfluidics is used in multiple applications, such as serology/immunology, separation [13,14], centrifugal platform/blood analysis, cell sorting, and organ-on-a-chip [15]. Standard microfluidic methods for manipulating cells include electrophoresis, dielectrophoresis, magnetophoresis, pillar-based separation method, thermophoresis, inertial forces, optical tweezing, and acoustophoresis [16].

A promising candidate matching the requirements for preclinical and clinical in vitro cell tests is the combination of acoustophoresis/acoustofluidic, which is the purely-electro-mechanical and label-free manipulation of cells and particles, which adds additional functionalities for lab-on-a-chip technologies, as introduced by Andreas Manz [17]. However, this approach is limited, as only one cell per trap can be analyzed. Lim et al. [18] reported cell assembly and aggregation, demonstrating the creation of erythrocyte aggregates via acoustic levitation, while Wu et al. [19] showed the acoustic assembly of cell spheroids in disposable capillaries. Iranmanesh et al. [20] designed an acoustofluidic device to achieve cell alignment, separation, and trapping. The system employs hydrodynamic flow focusing, and utilizes three piezo transducers to perform specific functions. To avoid interference due to multiple resonant zones, they have used ultra-low flow rates, reducing the device’s throughput and, thus, increasing the response time. In the theoretical analysis, the authors demonstrated trapping at multiple nodes in the z-plane. However, their experimental observations revealed that cells are trapped stochastically, forming cell aggregations within a flat chamber more related to 2D cell cultures. Svennebring et al. [21] showed a confocal ultrasonic cavity-based acoustic device for bioparticle selection and retention. This device showed multi-node trapping positions, which makes it hard to analyze cell–cell interaction due to the stochastic aggregates formed in the nodes. These microfluidic or acoustofluidic systems are often limited to the cells attached to a surface, making it possible to analyze cells only in 2D [20,21]. Athanasia et al. [22] demonstrated an on-chip 3D cell culture by ultrasonic standing waves in a multi-well microplate. They cultured HepG2 and adherent cells for two to seven days, requiring pipetting steps to replace the cell culture media. While this chip enables 3D culturing, microscopy analysis is diffraction-limited and requires fluorescence labeling. Park et al., 2023 [23] used an acoustic-based bioreactor, showing the hanging drop method for spheroid fabrication in acoustic levitation. They cultured human mesenchymal stem cells (MSCs) in only 35 µL of media without any continuous flow of nutrients for 12 or 24 h. In an interesting article from Luo et al., 2022 [24], the authors demonstrated a portable integrated acoustic platform for 3D cell culturing requiring only 1 µL of sample. The cell sample was filled using capillary force. They cultured for several days without any continuous supply of nutrients, making the system unfavorable for clinical applications. Hu et al., 2022 [25] showed the construction of 3D cell patterning in a hydrogel sheet by placing six interdigitated transducers (IDTs) on a lithium niobite substrate. Different types of multi-layer cellular structure were formed using acoustics. However, a continuous acoustic was not applied, and once the 3D structure was formed, the acoustic was turned off and the cells were placed in incubator for several days. As a result, most of the studies overlooked the vital step of cell culture, i.e., the continuous perfusion of nutrients, which results in incompatible with clinical settings. To analyze human cell function, the nutrient supply must be in continuous perfusion to maintain cell functionality for several days, whereas in many articles, researchers have provided manual feeding of nutrients to the cells or placed the whole assembly in the incubator [19,22]. Researchers have demonstrated the stochastic patterning of cells inside the cavity under acoustics, which is not a practical method for clinical outcome to analyze the results of the cell function test due to unrepeatable nature of the experimental setup.

With this work, we aim to address the highlighted challenges of workflow solutions and shortcomings. Our acoustofluidics platform is focused on establishing a label-free 3D modelling system without cell-wall contacts, which allows a continuous perfusion workflow over several days, making it robust and fast. With continuous temperature and flow control, we can perform perfusion experiments of trapped cell aggregates under sterile conditions throughout for any specified duration. It opens up the possibility of 3D micro-cell–cell interaction studies that go far beyond the state-of-the-art due to stable positioning, no need for refocusing, a single trapping site, ability to trap from a single cell to several hundred cells, and repeatability. In addition, this platform can also be easily integrated with other technologies.

## 2. Materials and Methods

### 2.1. COMSOL Simulation

The simulation aims to model and analyze the laminar flow, acoustic pressure at the eigenfrequency, particle flow under acoustics, and fluidic flow. Thus, we have used three physics modules in COMSOL Multiphysics 6.1 for our design: laminar flow, acoustic pressure module, and particle tracing module.

### 2.2. Prototype Development

The prototype development initiates with the assembly of the 3D-printed chip carrier, incorporating a glass chip featuring a spherical cavity (IMT, Masken und Teilungen GmbH, Greifensee, Switzerland), as depicted in Figure 1A,B. Electronic components such as a PZT transducer (2 MHz working frequency, Meggitt, Kvistgård, Denmark), a temperature sensor Pt100, and a Peltier element are integrated into the assembly. This study used a 70 mm long glass chip with an integrated wet-etched 500 µm diameter micro-spherical cavity and a channel width of 150 µm. First, the Peltier element is placed on the cooling fins and stuck with double-sided thermal tape. Next, a PZT is glued to the Peltier element; the temperature sensor Pt100 is glued on top of the transducer, as shown in Figure 1B. Lastly, the microfluidic chip is glued to the PZT transducer based on the developed prototype [26,27]. 

The setup utilizes a pressure pump from Biophysical Tools (Biophysical Tools GmbH, Wettin-Löbejün, Germany) and a temperature controller from Belektronig (BelektroniG GmbH, Freital, Germany). The experiment was carried out using a Leica fluorescence microscope with an Oku Lab incubator. The temperature in the incubator was set at 37 °C and the Peltier at 33 °C. An air supply was installed for the piezo cooling.

### 2.3. Cell Culture and Experimental Conditions

The human lymphoblast K-562 cell line (ATCC CCL-243) is used for the cell proliferation study. This cell line grows in suspension and is commonly used as a tumor target cell for in vitro experiments with natural killer cells. The suspended K-567 cells were thawed and cultured in the complete medium (consisting of Dulbecco’s Modified Eagle Medium (DMEM), 5% fetal calf serum (FCS), 4.5 g/L glucose amino acids, vitamins, salts, a buffer, and an oxygen supply used for cell culture). The TC20 (BIO-RAD, Hercules, CA, USA) was utilized to count the cultured cells.

## 3. Results

### 3.1. Background Theory and Numerical Solution

The Helmholtz equation is a partial differential equation that governs wave propagation. It can describe the equation of acoustic waves in a spherical cavity [28]. For a scalar acoustic field P in a spherical coordinate system, the Helmholtz equation is given by
(1)$$\nabla^{2}P+k^{2}P=0,$$
where k is the wave number and $\nabla^{2}$ the Laplacian operator in spherical coordinates. The spherical coordinates are typically denoted as $\left(r,\theta ,\emptyset \right)$, where r is the radial distance from the center of the sphere, θ is the polar angle measured from the *z*-axis, and ∅ is the azimuthal angle measured from the *x*-axis in the x–y plane. 

The complete solution for the acoustic wave inside a spherical cavity is then given by the product of the radial and angular parts:(2)$$P\left(r,\theta ,\emptyset \right)=R\left(r\right)\cdot \theta \left(\theta \right)\cdot \emptyset \left(\emptyset \right)$$

The eigenfrequencies $f_{n,l,m}$ of the acoustic modes inside the spherical cavity are defined by the boundary conditions at the surface of the sphere.

Where $R\left(r\right)$ is the radial part which depends on $\left(n,l\right)$ and $\theta \left(\theta \right)\cdot \emptyset \left(\emptyset \right)$ is the angular part. Further Equation (2) may be written as [29]:(3)$$P_{nlm}\left(r,\theta ,\emptyset \right)=R_{n}\left(r\right)\cdot Y_{lm}\left(\theta ,\emptyset \right)$$

The angular part is also described as spherical harmonics $Y_{lm}\left(\theta ,\emptyset \right)$, where the values of l, m depend on the mode of the spherical cavity.
(4)$$Y_{lm}\left(\theta ,\emptyset \right)=\theta_{lm}\left(\theta \right)\cdot \emptyset_{m}\left(\emptyset \right)$$
(5)$$\emptyset_{m}\left(\emptyset \right)=\frac{1}{\sqrt{2\pi }}e^{im\emptyset }$$

$e^{im\emptyset }$ has a dependency on azimuthal angle (∅), and also provides the azimuthal symmetry.
(6)$$\theta_{lm}=\epsilon \sqrt{\frac{2l+1}{2\pi }\frac{\left(l-\left|m\right|\right)!}{\left(l+\left\lceil m\right\rceil \right)!}}P_{l}^{m}\cos\theta$$
where $P_{l}^{m}\cos\theta$ is the Legendre polynomial of the order m and the degree l and which is related by −l≤m≤l depending on the polar angle (θ). The constant ϵ is 1 if m≤0 and ${\left(-1\right)}^{m}$ if m>0. Substituting Equations (5) and (6) into Equation (4) gives the angular part as follows:(7)$$Y_{lm}\left(\theta ,\emptyset \right)=\frac{\epsilon }{\sqrt{2\pi }}\sqrt{\frac{2l+1}{2\pi }\frac{\left(l-\left|m\right|\right)!}{\left(l+\left\lceil m\right\rceil \right)!}}P_{l}^{m}\cos\theta \cdot e^{im\emptyset }$$

The radial part is expressed in terms of the spherical Bessel functions:(8)$$R\left(r\right)=j_{n}\left(kr\right)+y_{n}\left(kr\right),$$
where $j_{n}\left(kr\right)$ and $y_{n}\left(kr\right)$ are the first and second kind spherical Bessel functions of order n, respectively. At origin, $y_{n}\left(kr\right)$ does not play a role in the acoustic field inside the spherical cavity. Thus, the radial part can be represented by $R\left(r\right)=j_{n}\left(kr\right)$. 

As mentioned above, in this study, we have used the glass chip with a spherical cavity of approximately 500 µm diameter filled with water. The calculated lowest eigenfrequency equates 1.963 MHz, which corresponds to the degenerate pair of modes (1,1,m). Thus, the angular part (for all the cases l=1) is specified as follows:

If m=−1 then the solution of $Y_{lm}\left(\theta ,\emptyset \right)$ leads to
(9)$$Y_{1}^{-1}\left(\theta ,\vartheta \right)=\sqrt{\frac{3}{8\pi }}\cos\varphi \sin\theta -i\sqrt{\frac{3}{8\pi }}\sin\varphi \sin\theta$$

If m=0 then the solution of $Y_{lm}\left(\theta ,\emptyset \right)$ leads to
(10)$$Y_{1}^{0}\left(\theta ,\vartheta \right)=\sqrt{\frac{3}{4\pi }}\cos\theta$$

Equation (10) shows that the angular part $Y_{1}^{0}$ is purely real and does not depend on the azimuthal angle ∅. This implies that the spherical harmonic looks the same in all directions within the horizontal plane. 

If m=+1 then the solution of $Y_{lm}\left(\theta ,\emptyset \right)$ becomes
(11)$$Y_{1}^{1}\left(\theta ,\vartheta \right)=-\sqrt{\frac{3}{8\pi }}\cos\varphi \sin\theta -i\sqrt{\frac{3}{8\pi }}\sin\varphi \sin\theta$$

For *n* = 1, the spherical Bessel function is given by
(12)$$j_{n}\left(kr\right)=\frac{1}{k_{nl}r}\sin\left(k_{nl}r-\frac{\pi }{2}\right)$$

The eigenfrequencies $f_{nl}$ of the acoustic modes inside the spherical cavity is defined by [30]
(13)$$f_{nl}=\frac{ck_{ln}}{2\pi }$$

Due to the azimuthal angle (∅) degeneracy, the eigenfrequency remains consistent, irrespective of variations in the azimuthal angle. It therefore does not depend on m.

Further, 3D numerical simulations were performed with COMSOL Multiphysics to determine the eigenvalues and to simulate particle trapping under acoustic radiation force at mode 1 (1,1,m). The acoustic pressure P shown in Figure 2A–C, with dark blue, red, and green colors indicating negative, positive, and zero pressure (node), respectively. As shown in Figure 2D–F, the larger particles of 10 µm, which do not influence acoustic streaming force, move towards the node (center of spherical cavity) due to the acoustic radiation force where the acoustic pressure is zero (Figure 2F). Consequently, single trap was achieved, which helps in trapping of cells in non-stochastic pattern. This considerably enhances the cell growth analyses and cell–cell interactions in a wide range of pre-clinical application.

### 3.2. Cell Biology

The cell line K562 was used to validate the acoustic-based cell trapping system. At the start of the experiment, the cell density in a cell culture flask was counted with a TC20 Cell Counter (Bio-Rad). The cell density was 2 × 10^6^ cells/mL (excluding the dead cells, as detected from Trypan Blue dye). Since 72 h before the cell count was 0.6 × 10^6^ cells/mL, the doubling time of the culture is approximately ≈40 h. Accordingly, strict compliance regarding a fixed cell culture protocol (e.g., growth rate, cell number) was considered for the trapping experiment.

### 3.3. Experimental Detail

Initially, the inlet pressure was kept at 150 mbar. Once the cells started flowing inside the microfluidic glass chip, the fluidic pressure was adjusted to 0 mbar, and the acoustics were applied using an electrical setup. A sweep refers to a recurring, continuous variation of frequency within a specified range, accompanied by an alternating voltage of consistent magnitude. The duration of the sweep was 70 ms. For mode 1, which was used for the following experiments, the sweep was between 1.75 MHz (start value) and 1.85 MHz (end value), depending on the number and size of the cells and the prototype glass channel. The end value was optimized around 1.9 to 2.0 MHz. The voltage set on the function generator was 5 V, but it was amplified to ~15 V within the circuit. In the beginning, five cells were acoustically trapped in mode 1 at defined electrical parameters, indicated in Table 1. 

The study was planned to continuously observe cell growth for 55 h, and an image was taken every five minutes. Furthermore, after trapping the cells at the center of the spherical cavity (node), a background medium flow for maintaining cell culture conditions was adjusted at a pressure of 2 mbar. This pressure is a tradeoff between constant medium supply and the prevention of shear stress. In addition, the incubator’s temperature was set at 37 °C, and the temperature of the Peltier element was kept at 33 °C to avoid temperature gradients and keep the temperature in the cell trap at 37 °C.

## 4. Discussion

A significant outcome of this work was achieved with the cell proliferation experiment performed with K562 cells, as mentioned in the previous section. Initially, five cells were trapped at the center of the spherical cavity within one minute, as shown in Figure 3A. The single trap was attained, as explained in the previous section. There was no need to refocus for several days, making the system robust for cell function analysis and continuous perfusion study. The progression of cell division is shown in Figure 3. Approximately 12 h later, one of these cells underwent cell death and debris formation (Figure 3B). The first cell division happened after about 33 h with five trapped cells again (Figure 3E). After ~43 h, two of the five cells died, as indicated in Figure 3F. Within the following 2 h, one more cell died, but cell division was observed again after another 20 h (Figure 3G–I). Notably, a cluster of cell debris was observed in the background or adjacent to the trapped cells. Furthermore, the 2nd cell division was observed after 53 h (Figure 3I,J). 

One point that must be emphasized is the cells’ condition at the experiment’s beginning. Since a small number of cells were trapped from a large population, i.e., 0.6 × 10^6^ cells/mL, cell death was expected. Additionally, cells are not synchronized and may be in distinct growth phases. Accordingly, the cell viability experiment (55 h) under acoustic trapping was also conducted to demonstrate a safe, cell-preserving workflow (Appendix A).

A mitotic event can be observed in Figure 3, where (Figure 3D,G) shows the process of cytokinesis. 

The cell undergoes rotation around the *z*-axis of the bright-field microscopy before stabilizing its position and subsequently dividing into two daughter cells. Intriguingly, a physical connection between the cancer cells persists even post-division, as evidenced in Figure 3E,H, where the daughter cells take on a configuration resembling the numeral “8”.

Over time, the daughter cells are further strangulated into two separate daughter cells until they are no longer connected and form two independent cells, as shown in Figure 3J. This process is very similar to the mitosis process described by Mao and Yin [31]. When a cell divides, it can be observed how one cell becomes two daughter cells. Additionally, cell rotation was observed after the formation of two daughter cells, as shown in Figure 3I,J. Cell rotation could be observed 30 to 60 min before and during the cell division.

When there is a substantial shift in the mass and density of individual cells or cell clusters, rotation or movement of the cells is generally anticipated. Consequently, variations in the number of trapped cells lead to variations in the acoustic radiation force (ARF). Cell rotation is influenced by ARF variation, which is applied to the particles and produces torque. The acoustic radiation force is directly proportional to the volume of the particles or cells and the contrast factor, as shown in Equations (14) and (15). The contrast factor is a function of particle/cell and fluid density. Thus, the higher the contrast factor, the better the acoustic radiation force; therefore, more dense particles can be trapped. This can affect the trapping efficiency or confinement of acoustic waves within the device [32].
(14)$$F_{Ax}=4\pi R^{3}Ek\sin\left(2kx\right)\emptyset$$
(15)$$\emptyset =\frac{\rho_{P}+(2/3)\left(\rho_{P}-\rho_{0}\right)}{2\rho_{P}+\rho_{0}}-\frac{1}{3}\frac{\rho_{0}c_{0}^{2}}{\rho_{P}c_{P}^{2}}$$
where $F_{Ax},\;E,\;R,\;x,\;\rho_{P}$,$\;\rho_{0}$, $\rho_{0}$, $c_{P}$, $c_{0}$, ∅, are acoustic radiation force, acoustic energy density, radius of the cell or particle, cell position in the direction of wave propagation, density of cells/particles, density of liquid, and contrast factor, respectively. $k=2\pi f/c_{0}$ where f is frequency. 

The density of the cells or buffer also gives information about the acoustic impedance. In order to eliminate the reflections or acoustic losses, the acoustic impedance should be matched for both media, which will improve the trapping efficiently. The relationship between acoustic impedance (Z) is defined by the density (ρ) and the speed of sound (c):(16)Z=ρc

To achieve maximum value of acoustic energy density in the buffer, the system is designed so that the cavity wall material has a lower acoustic impedance than the transducer, but higher acoustic impedance than the fluid in the cavity.

The rotation was observed in the xy plane, which indicates that the rotation is primarily influenced by the azimuthal angle (∅). The rotation was also observed during the cell decomposition for about 60 min (Appendix A). In this case, the acoustic trap is expected to be stable. This is also supported by the fact that in both cases, the cells remained stationary for hours, and no movement was observed before rotation started. The cells lasted more than half an hour, leading to the completion of cell division. These results match with earlier work [31,33], which claims that the migration of a cell ends and that it gets stable before it divides. The documented slight rotation during the cell division process, as highlighted in Figure 3I,J, is following the typical processes described by authors during mitosis. First, the cell’s movement speed decreases. Further, the size of the cell decreases, and its brightness increases, forming an “8” during the cell division process. This recurrent process is also detailed in other publications, including the study by Huh et al. [33].

Some experiments were performed to check the viability of the cells using PI staining. During the viability experiment, only a few cells died among other trapped cells, which was carried out for 55 h, as shown in Appendix A. The calculated live and death rates were ~95% and ~5%, respectively (see Appendix A). Nutrients were continuously supplied for 55 h at pressure of 2 mbar, and the incubator temperature was also controlled to 37 °C. 

The possible factors of cell death during the experiment are the acoustic radiation force, the shear stress induced by the flow rate, and the condition of cells. However, a shear stress of ~0.07 dynes/cm^2^ was calculated at the center of spherical cavity, while for the whole cavity it was 0.2 dynes/cm^2^ using COMSOL FEM model, generated by nutrient flow at 2 mbar, as indicated in Figure 4. However, this value is far below the shear stress value in arteries and veins, i.e., 10–70 dynes/cm^2^ and 1–6 dynes/cm^2^, respectively [34,35]. The applied acoustic force was in the nN range. Thus, shear stress and the force generated by acoustic radiation do not cause cell death. Another important reason for cell death is the condition of the cells. These cells were cultivated as a cell culture according to a predefined protocol. The cells used in this experiment were assumed to be in a logarithmic growth phase in the cell culture at the start of the experiment. The cells in our acoustic trap probably died naturally, as many cells also died in conventional 2D cell Petri-dish-based cell culture. One indication of this is that the cells were all from different states and at different times in their life cycle.

## 5. Conclusions

Our acoustic platform is the integration of a 3D acoustic levitation assembly with microfluidic systems or lab-on-a-chip platforms, which can highlight its potential for enhancing functionalities in various micromachine applications. It enables the acquisition and visualization of experiments with single to hundreds of cells or cell aggregates trapped over extended periods. The simulations conducted in this study align closely with the experimental observations. The stability of the trap not only ensures a stable position of the cells over a single trap in a non-stochastic manner, but also that no microscopic refocusing is necessary. Trapped micro aggregates can be studied under perfusion conditions, allowing effortless optical monitoring of cell proliferation. In addition, unlike most platforms that require cell adhesion to a surface, our acoustofluidic platform enables the 3D analysis of cells, as they do not require surface attachment, while still allowing for permanent microscopic visualization without the need for refocusing. Coupled with the capacity for continuous perfusion and 3D visualization, our acoustofluidic platform is a promising tool for cell function studies. The detected cell divisions during acoustic trapping affirm cell viability preserving conditions.

## Figures and Tables

**Figure 1 micromachines-15-00436-f001:**
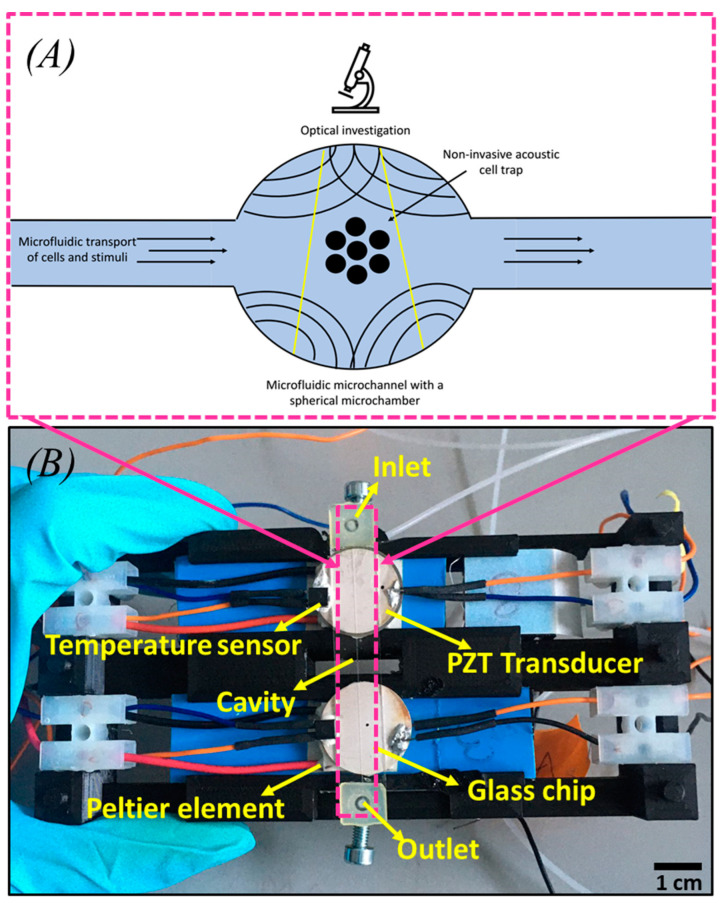
(**A**) Schematic of trapped cells in a microfluidic glass chip under acoustic levitation. (**B**) Prototype for acoustic trapping and temperature control.

**Figure 2 micromachines-15-00436-f002:**
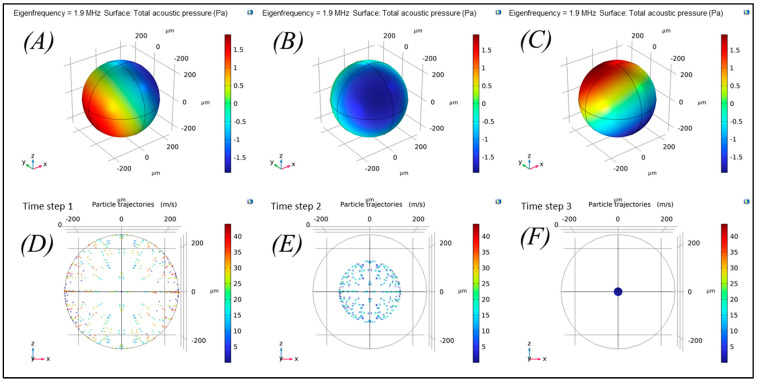
(**A**–**C**) Acoustic pressure field at mode 1 (1,1,*m*), simulated in COMSOL Multiphysics; (**D**–**F**) trapping of 10 µm particle at the node (center of the spherical cavity) under acoustic radiation force.

**Figure 3 micromachines-15-00436-f003:**
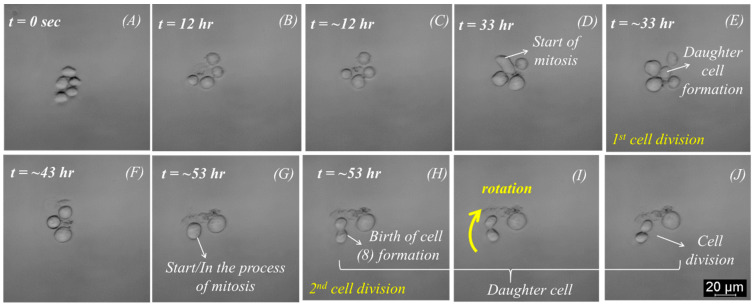
Suspended K562 cells were trapped at mode 1 (1.75 MHz–2.00 MHz)—(**A**) initially, five cells were trapped; (**B**,**C**) one of the trapped cells decomposed; (**D**,**E**) start of mitosis and 1st cell division; (**F**–**H**) cell death; (**I**,**J**) 2nd cell division with a pronounced rotation around the optical *z*-axis.

**Figure 4 micromachines-15-00436-f004:**
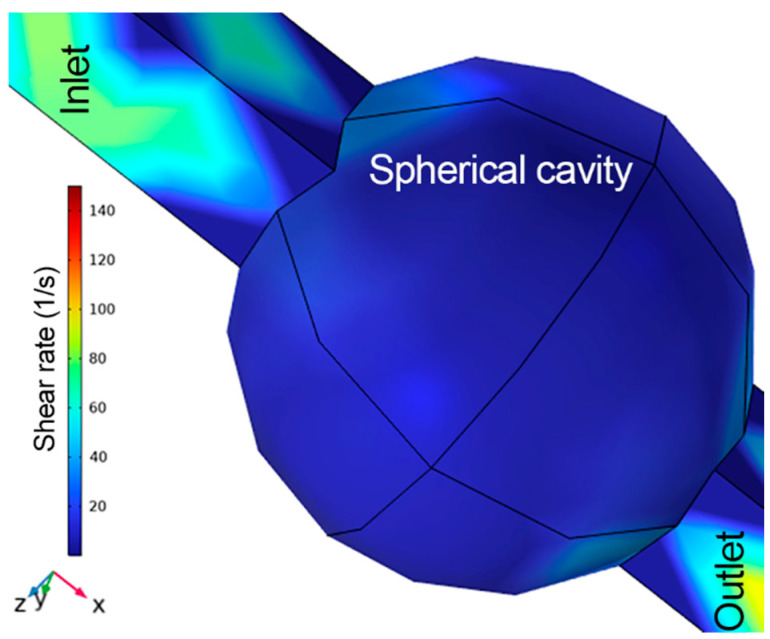
Shear rate (1/s) profile at flow rate of 2 mbar, simulated in COMSOL Multiphysics, which shows the shear rate inside spherical cavity lies below 20/s (shear stress: ~0.2 dynes/cm^2^).

**Table 1 micromachines-15-00436-t001:** Electrical and fluidic parameters for cell proliferation experiment.

Parameters	Values
Sweep (ms)	70
Frequency (MHz)	1.75–2.0
Voltage (Vpp)	5.0
Initial fluidic pressure (mbar)	150
Fluidic pressure of media (mbar)	2

## Data Availability

The data that support the findings of this study are available from the corresponding author upon reasonable request.

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
