# Peer review of "Continuous Perfusion Experiments on 3D Cell Proliferation in Acoustic Levitation"

_micromachines, 2024, doi:10.3390/mi15040436_

Round 1
Reviewer 1 Report
Comments and Suggestions for Authors
Fabiano et al. reported an ultrasonic microfluidic device for trapping and culturing K562 cells. Cells were trapped at the center of a spherical cavity inside a glass microdevice by the standing acoustic wave field, and long-term observation was achieved using a microscope. The proliferation of the K562 cells over a period of >50 hr was demonstrated. However, I cannot recommend the publication of this work at the current stage due to the lack of significance and the concerns regarding the experimental results.
1. There are numerous reports using standing acoustic field to trap and culture cells. It is unclear what new results the present work will bring to the field are. Authors stated that “address the highlighted challenges of workflow solutions and shortcomings in analysis” What are the “highlighted challenges of workflow solutions”? The present work used optical microscopy for analysis, which is also a standard method for other acoustofluidic systems.
2. Nutrients were delivered using a pressure of 2 mBar. How much shear stress will be applied to the cells under this pressure?
3. Statistical analysis of the cell culture results is lacking. While cell dividing was indeed observed, significant numbers of cells also died during this process. The negative impact of the present system on living cells needs to be carefully evaluated.
4. In addition, the present application does not support the significance of the single cavity system. Compared with the single trapping system, arrayed trapping systems (like the ones mentioned in the introduction) are clearly advantageous as they can process large amount of cells providing statistically significant results in one experiment run.
Reviewer 2 Report
Comments and Suggestions for Authors
This work investigates a useful tool to accomplish rational, non-invasive manipulation and positioning of microparticles and cells in the spherical cavity. Captured particles and cells can be studied in perfusion conditions, which provides an effective method in effortless optical monitoring of cell proliferation. So, I think this interesting and meaningful work can be publishable after major revision.
(1) Scale bar should be added in Fig.1B, and Fig.3A-J.
(2) How does the cell density affect the cell trapping results?
(3) References from the past three years should be added.
(4) Possible factors causing cell death in Fig.S1 should be illustrated in the manuscript.
(5) Advantages of this method should be demonstrated in the manuscript.
Comments on the Quality of English LanguageMinor editing of English language required
Round 2
Reviewer 1 Report
Comments and Suggestions for Authors
Most of my concerns have been addressed. More details regarding the cell viability experiment should be provided. How many cells were used for the calculating the 95% viability rate? How was the experiment conducted? Are the results from a single device or multiple devices? How viability was measured, especially for 3D cultures?
Author Response
File attached

Reviewer 2 Report
Comments and Suggestions for Authors
I think this work is publishable in Micromachines.
Author Response
We would like to thank you for the valuable suggestions.
Round 3
Reviewer 1 Report
Comments and Suggestions for Authors
my concerns have been addressed.
Author Response
We would like to thank the reviewer for the valuable suggestions and comments.